# Degenerated Virulence and Irregular Development of *Fusarium oxysporum* f. sp. *niveum* Induced by Successive Subculture

**DOI:** 10.3390/jof6040382

**Published:** 2020-12-21

**Authors:** Tao-Ho Chang, Ying-Hong Lin, Yu-Ling Wan, Kan-Shu Chen, Jenn-Wen Huang, Pi-Fang Linda Chang

**Affiliations:** 1Department of Plant Pathology, National Chung Hsing University, Taichung City 402204, Taiwan; taudch@gmail.com (T.-H.C.); yonea1026@gmail.com (Y.-L.W.); jwhuang@nchu.edu.tw (J.-W.H.); 2Innovation and Development Center of Sustainable Agriculture (IDCSA), National Chung Hsing University, Taichung City 402204, Taiwan; 3Department of Plant Medicine, National Pingtung University of Science and Technology, Pingtung 912301, Taiwan; pmyhlin@mail.npust.edu.tw; 4Plant Medicine Teaching Hospital, General Research Service Center, National Pingtung University of Science and Technology, Pingtung 912301, Taiwan; 5Chiayi Experiment Branch, Taiwan Agricultural Research Institute, Chiayi 611002, Taiwan; kschen@dns.caes.gov.tw

**Keywords:** *Fusarium**oxysporum* f. sp. *niveum*, degenerated variants, pathogenicity, virulent factors, watermelon Fusarium wilt, cell wall-degrading enzymes

## Abstract

Successive cultivation of fungi on artificial media has been reported to cause the sectorization, which leads to degeneration of developmental phenotype, and virulence. *Fusarium oxysporum* f. sp. *niveum* (Fon), the causal agent of watermelon Fusarium wilt, forms degenerated sectors after successive cultivation. In the present research, we demonstrated that subculture with aged mycelia increased the incidence of degenerations. To further investigate the differences between the Fon wild type (sporodochial type, ST) and variants (MT: mycelial type and PT: pionnotal type), developmental phenotypes and pathogenicity to watermelon were examined. Results in variants (PT2, PT3, PT11, and MT6) were different from ST with mycelia growth, conidia production and chlamydospore formation. Virulence of degenerated variants on susceptible watermelon Grand Baby (GB) cultivar was determined after inoculation with Fon variants and Fon ST. In root dipping methods, Fon variants showed no significant differences in disease progress compared with ST. Fon variants showed a significant decrease in disease progression compared with ST through infested soil inoculation. The contrasting results of two inoculation methods suggest that the degenerative changes due to repeated successive cultivation may lead to the loss of pathogen virulence-related factors of the early stage of Fon infection process. Therefore, cell wall-degrading enzymes (CWDEs; cellulase, pectinase, and xylanase) activities of different variants were analyzed. All Fon degenerated variants demonstrated significant decreases of CWDEs activities compared with ST. Additionally, transcript levels of 9 virulence-related genes (*fmk1*, *fgb1*, *pacC*, *xlnR*, *pl1*, *rho1*, *gas1*, *wc1*, and *fow1*) were assessed in normal state. The degenerated variants demonstrated a significantly low level of tested virulence-related gene transcripts except for *fmk1*, *xlnR, *and *fow1*. In summary, the degeneration of Fon is triggered with successive subculture through aged mycelia. The degeneration showed significant impacts on virulence to watermelon, which was correlated with the reduction of CWDEs activities and declining expression of a set of virulence-related genes.

## 1. Introduction

*Fusarium oxysporum *(Fo) is a soil-borne plant pathogen causing a devastating effect on agricultural crops worldwide. The pathogen is very plastic and encompasses several formae speciales according to their host range [1]. Fusarium wilt of watermelon, caused by *F. oxysporum *f. sp. *niveum *(Fon), is one of the limiting factors for watermelon production worldwide [2,3]. During most of watermelon growing stages, Fon infects watermelon and causes watermelon production a loss of 10% to 15% of yield [4,5]. One of the most recommended methods for controlling this disease is via breeding program for resistant lines; however, this is a time-consuming approach [2]. Another approach is using non-pathogenic *F. oxysporum*, which antagonizes the pathogenic Fo or primes host immune response. Therefore, understanding the pathogenesis and virulence of Fo is pivotal in controlling Fusarium wilt in the field [6].

The progress of Fo established in their host contains four significant steps [7]. First of all, Fo requires the signals of host plants to start their invasion processes. Roots secret a broad range of organic signals such as sugars and amino acids which can trigger Fo spore germination and support germ-tube growth [8]. Secondly, germ tubes of Fo will then grow to root surfaces where the first contact point of pathogen–plant interaction begins [9]. Fo attaches to the surface of root hairs and forms a mycelial network and colonizes the host root system [10]. Thirdly, Fo invades root cortex and vascular tissues and differentiates within xylem vessels. Fo secrets a broad range of enzymes to breakdown the first layer of basal plant defense [11]. Finally, Fo secrets toxins and virulence factors to assist disease evolvement in the host plant [12]. The virulence factors regulate the pathogenesis of Fo in host plants in every steps of Fusarium wilt progress.

Degeneration is a common feature for most fungi in culture on artificial media. This commonly happens through repeated successive subculture [13]. These changes include phenotypic degeneration and virulence attenuation. Additionally, successive subcultural degeneration will affect fungal pigmentation, growth, morphology, spore production, changes in metabolic products, and production of variant sectors [13]. Studies have demonstrated the occurrence of phenotype degeneration of *Fusarium *spp. after successive subculture [14]. In the normal state, wild-type Fo grows on media by stacking the macroconidia and forming sporodochia [15]. Reports have showed there were two types of Fo colonies forming after degeneration, mycelial type (MT) and pionnotal type (PT) [16]. The mycelial type of Fo produces mycelia without pigment deposition and conidia production. In contrast, the pionnotal type produces a reduced amount of mycelia but with a massive amount of conidia spores forming slimy type of colonies. The development of fungi after degeneration has been described; however, the impact of degeneration on virulence remains unclear.

Although a number of studies have addressed the phenomenon of fungi degeneration, the major impact of this process to virulence of *F. oxysporum* remains unclear. Therefore, we choosed to investigate the impacts of fungal degeneration caused by successive subculture using *F. oxysporum* f. sp. *niveum* (Fon-H0103). First of all, we replicated successive cultures for ten generations and observed the occurrences of degeneration at various generations. Secondly, we selected four different degenerated variants for the following analyses to identify impacts of degeneration to development and virulence of Fon. Lastly, we examined the cell wall-degrading enzyme activities and virulence-related gene expressions to reveal the mechanisms of fungal degeneration.

## 2. Materials and Methods

### 2.1. Pathogen Resources and Cultivation

*Fusarium oxysporum *f. sp. *niveum* (Fon-H0103 isolate) was grown on 1/2 PDA (half-strength potato dextrose agar, potato extract 200 g L^−1^, 1% D-glucose, 2% agar). Four degenerated variants (PT2, PT3, PT11, and MT6) were derived from successive subcultures of Fon-H0103 sporodochial type (ST). Fon ST and degenerated variants were maintained with single spore culture every two weeks on Nash-PCNB plate (1.5% peptone, 2% agar, 0.1% KH_2_PO_4_, 0.5% MgSO_4_·7H_2_0, 0.1% pentachloronitrobenzene, 0.03% streptomycin and 0.1% neomycin) to avoid further degeneration [17].

### 2.2. Plant Materials and Growth

Susceptible watermelon (Grand Baby, (GB), Know-You Seed Co. Ltd., Kaohsiung, Taiwan) was used in this research. GB seeds were treated with running water at room temperature for two days. The germinated seeds were placed on water-soaked filter papers (ADVANTEC^®^, Tokyo Roshi Kaisha, Ltd., Tokyo, Japan) in the plastic Petri dish for two days in the dark. The cultivated GB seedlings were then subjected to the pathogen inoculation.

### 2.3. Generation of F. oxysporum Degenerated Variants

*F. oxysporum *f. sp. *niveum* (Fon ST) were cultured on half-strength PDA for one week. The agar plugs from the aged mycelia and hyphal tip, as shown in the scheme (Appendix A), were used for successive subculture. At least ten plugs of Fon mycelia and hyphal tip were used in each subculture procedure. The subculture process was performed for 10 generations. The degenerated or sectorization of Fon was then isolated as a single spore to maintain the degenerated variants. The pattern of fungal degeneration was recorded, and the transformation rate was calculated. In the end, three pionnotal types (PT2, PT3, and PT11) and one mycelial type (MT6) were selected for further experiments.

### 2.4. Phenotyping of Fungal Growth

Growth phenotypes of Fon ST and degenerated variants (PT2, PT3, PT11, and MT6) were examined by three types of growth indices: mycelia growth, spore production, and chlamydospore formation. For mycelia growth, the single spore of Fon ST and degenerated variants were placed in the center of half-strength PDA plate. The colony diameter was measured after a 7-day incubation at 28 °C. To elucidate the ability of conidium production, 0.5 cm^2^ colony plugs from 3-week old colonies of Fon ST and degenerated variants were cut from half-strength PDA and washed in 2 mL sterilized water to make spore suspensions. Spore suspensions were then examined under a microscope and the number of conidia was calculated by hemocytometer. Chlamydospore formation of each variant was determined according to the method described by Awuah and Lorbeer (1988) with minor modification [18]. The soil extract solution (500 mL) was prepared for inducing the chlamydospore formation by shaking 50 g of organic soil (peat moss soil BVB No.4: Bas Van Buuren Co., Ltd., Maasland, South Holland, The Netherlands) and 500 mL sterilized dH_2_O for one hour. The slurry was filtrated through a 9-cm Whatman No. 1 filter paper for clarification of the soil extract. The soil extract was sterilized by filtering through a 0.2 µm Millipore filter (Millipore Corp., Bedford, MA, USA). Five hundred µL of conidia suspension (10^4^ conidia mL^−1^) was mixed with 6 mL of sterilized soil extracts and further plated on a 5-cm Petri dish at room temperature for seven days. The chlamydospore formation was observed and counted by using glass hemocytometer under a microscope.

### 2.5. Methods for Pathogen Inoculation

Fon ST and degenerated variants (PT2, PT3, PT11, and MT6) were used for inoculation tests to determine their virulence or pathogenicity. Two inoculation methods were applied. For root dipping method, susceptible watermelon GB seedlings were grown on peat moss soil BVB No.4 (Bas Van Buuren Co., Ltd., Maasland, South Holland, The Netherlands). Two-week old GB seedlings were up-rooted and cleaned with running water to reveal the seedlings’ root system. The second method was using infested-soil for inoculation according to Chang et al. (2015) [2]. Fon ST and degenerated variants were cultivated in half-strength PDA, the two-week old Fon mycelia and spores were collected from the plate and mixed with sterilized sand. The concentration of Fon in infested-soil was determined by plating the serially diluted soil suspensions on PCNB plates and calculated the concentration of Fon in the stock infested soil. The stock infested-soil was then diluted with peat moss soil BVB No.4 to 10^4^ spores per gram of soil. Watermelon seedlings were grown in the soil infested with Fon ST and degenerated variants for inoculation.

### 2.6. Cell Wall-Degrading Enzymes Assay

Cell wall-degrading enzymes (CWDEs) were assayed according to the method modified by King et al. (2009) [19]. Different variants of Fon were cultured in Bilay and Joffe’s medium with modifications (0.1% KH_2_PO_4_, 0.1% KNO_3_, 0.05% MgSO_4_·7H_2_O, 0.05% KCl, 0.02% sucrose, 0.02% glucose, 0.15% carboxymethyl cellulose, 0.15% xylan, 0.15% pectin, pH 4.0) for 7 days, and the growth medium was collected for the analyses of the activities of cellulase, pectinase, and xylanase [20]. At the beginning of the assay, the filtered enzymes extracts from growth medium were subjected to Bradford’s protein assay to quantify the total protein concentration [21]. CWDEs’ enzyme activities were determined by the production of reducing group after enzyme reaction with the colorimetric assay by using 3,5-dinitrosalicylic acid (DNS). The reaction solution containing carboxymethylcellulose, xylan, and citric pectin was used for cellulase, xylanase, and pectinase activities, respectively, with minor adjustment (pH = 4.8). Activities of CWDEs were determined by analyzing the production of reducing sugar after an hour reaction at 40 °C.

### 2.7. Real-Time Quantitative Reverse Transcription PCR (qRT-PCR) of Virulence Gene Expression

For gene expression analysis, variants of Fon obtained from overnight PDB (potato extract 200 g L^−1^, 2% D-glucose) cultures were regrown for 12 h in freshly prepared PDB medium. Cultures of mycelia were filtered through three layers of MiraCloth (Calbiochem Corp., La Jolla, CA, USA) for nucleic acid isolation. Total RNA was extracted using the RNAzol^®^RT (Molecular Research Center, Inc., Cincinnati, OH, USA) according to the manufacturer’s procedure. Tested cDNA was prepared using the SuperScript™III RNaseH^−^ reverse transcriptase system (Invitrogen, Carlsbad, CA, USA) according to the manufacturer’s procedure. Real-time PCR was monitored on Rotor-Gene^®^ Q-Pure Detection System (Software Ver. 2, Qiagen Inc., Valencia, CA, USA) and performed using QuantiFast SYBR^®^ Green PCR Master Mix (Qiagen). Primers (Table 1) specific to each of the tested virulence-related genes were designed by NCBI (National Center for Biotechnology Information, Bethesda, MD, USA) net program Primer-BLAST. For real-time PCR assay, a 10-μL reaction mixture containing template cDNA (synthesized from 10 ng total RNA), each of 100 nM amplification primers, and 1X QuantiFast SYBR^®^ Green PCR Master Mix (Qiagen). The parameters for real-time PCR were set according to the procedure of the manufacturer (polymerase activation hold at 95 °C for 2 min, then 40 cycles of denaturing at 95 °C for 15 s and of annealing/extension for 1 min). After real-time PCR, melting curves (65 °C to 99 °C) of the PCR products were analyzed to verify the specificity of the amplified fragments. Three independent experiments were used for calculating the relative expression level of each virulence-related gene.

## 3. Results

### 3.1. Successive Subculture of Aged Mycelia Induces the Cultural Transformation

Results of successive culture from aged mycelia as illustrated in Figure 1 showed 10% transformation rate at first subculture and around 60% transformation rate at the 10th subculture. However, successive culture from hyphal tips showed no transformation during all subcultures.

### 3.2. Variants of Fon Demonstrated Significant Morphology Changes

Colony morphology of the Fon variants PT2 (Figure 2a), PT3 (Figure 2b), and PT11 (Figure 2c), variant MT6 (Figure 2d), and their parental Fon ST (Figure 2e) was pionnotal (Figure 2a–c ), mycelial (Figure 2d), and sporodochial (Figure 2e), respectively. When grown on PDA, the growth rates between the variants were not significantly different (Figure 3a). The production of conidia of all pionnotal variants was significantly higher than those of MT6 and ST-H0103 (Figure 3b), but the capacity of PT variants to produce chlamydospore was significantly lower than that of MT6 and ST-H0103 (Figure 3c). These data indicated that the tested Fon cultures had some cultural differences in conidiation and chlamydospore production but not in colony radial growth rate.

### 3.3. Variants of Fon Reduced Their Virulence to Susceptible Watermelon

Virulence of Fon variants was tested by root-dipping and infested-soil methods. Both the disease progress and severity of Fon variants were recorded (Appendix A). Virulence of variants was demonstrated by calculating the area under disease progress curve (AUDPC). The results of root-dipping showed that there was no significant difference in virulence between the Fon ST and other variants (Figure 4a). However, the results of infested-soil method demonstrated that virulence of Fon variants (PT2, PT3, PT11, and MT6) was lower than that of Fon ST (Figure 4b). With the two inoculation systems, we showed that Fon variants had lost their virulence as a consequence of degeneration.

### 3.4. Activites of Cell Wall-Degrading Enzymes (CWDEs) Were Reduced in Degenerated Variants

In order to prove the hypothesis that Fon variants may have lost their virulences due to the changes of their cell wall-degrading enzyme activity, a biochemical assay was performed. The enzyme assays showed that the activities of CWDEs cellulase (Figure 5a), pectinase (Figure 5b), and xylanase (Figure 5c) in Fon ST were all higher than those in the variants PTs and MT6. Cellulase and xylanase activities were close to zero which means ten times lower in variants than in Fon ST. Moreover, the pectinase activities of all degenerated variants were significantly lower than that of Fon ST. These results suggested that degeneration of Fon affected the CWDEs enzyme activity which might in turn reduce the virulence of Fon to watermelon.

### 3.5. Fungus Degeneration Affects the Expression of Virulence-Related Genes

Expression profiles of the virulence-related genes including *fmk1* (Figure 6a), *fgb1* (Figure 6b), *pacC* (Figure 6c), *xlnR* (Figure 6d), *pl1* (Figure 6e), *rho1* (Figure 6f), *gas1* (Figure 6g), *wc1*(Figure 6h), and *fow1* (Figure 6i) in the cultures of Fon were determined by qRT-PCR analysis. In comparison with ST-H0103, relatively low level of transcripts of the virulence-related genes (*fgb1*, *pacC*, *pl1*, *rho1*, *gas1* and *wc1*) were accumulated in mycelia of all variants. On the other hand, *fmk1*, *xlnR*, and *fow1* genes expression in all variants were not consistent. Expression of *fmk1 *was 2.2 times higher in PT3 than in ST, but 0.8 times lower in MT6 than in ST. PT11 showed no significant differences to ST in *xln*R and* fow1* genes expression and was significant higher than other variants. These observations indicate that a sufficient accumulation of the virulence-related factors in mycelia seems to be required for full pathogenesis of Fon.

## 4. Discussion

The degeneration is a common phenomenon in many fungi as a consequence of successive subculture on artificial media. Fo often shows morphological variation when cultured on carbohydrate-rich media, and frequently accompanies with the impairment in pathogenicity or virulence [15]. Degeneration in the subculture of *F. compactum* and *F. acuminatum *occurs more frequent when using single macroconidia spore for ten generations than using hyphal tips [14]. Interestingly, we have demonstrated that successive subculture of aged mycelia have higher transformation rate than that of hyphal tips. Our results suggest that the occurrence of degeneration may vary from different cultural stages of Fo.

The changes are usually reversible while fungus was grown back to the natural habitat instead of artificial environment [23]. However, previous research on Fon variants suggests direction of variation is irreversible when maintaining by single spore isolation [24]. Additionally, we have observed the morphology of variants remained the same while isolated from infested-soil. This suggests variants in present study are stable. Furthermore, these variants showed significant morphology differences compared with ST. The stable changes make these variants potential candidates for elucidating effects of cultural degeneration.

Our research demonstrated that the growth rates between Fon ST and degenerated variants on the PDA medium were similar. The observation in the growth rate of the morphological variants of Fon is consistent with the finding in *F. oxysporum* f. sp. *apii* [18] but not with that in *F. graminearum* [25]. Additionally, the production of conidia and formation of chlamydospore showed significant differences between Fon-H0103 ST and degenerated variants. These phenomena may affect the fitness and survival in vivo, which may affect the disease progress on watermelon.

Therefore, we assessed the virulence of Fon ST and degenerated variants. Our results showed that the degenerated variants delayed or attenuated the disease progress in watermelon with the infested-soil assay. However, the watermelon Fusarium wilt development of degenerated variants showed no significant differences with Fon ST through root-dipping method. The root-dipping method increases the opportunity for Fon to contact roots surfaces and the root system often is wounded through up-rooted methods [26]. Previous report suggests that broken or mechanically damaged roots may lead to wilt symptom in resistant crops [27]. In other words, the root dipping method has eliminated the first layer of plant defense system. Hence, we speculated that the degenerated variants might cause loss of virulence factors which were related to the stage of host-recognition and colonization.

Plant cell wall, the front layer for the plant defense system, is composed of polysaccharide-rich elements such as cellulose, hemicellulose, and pectin [28]. The cross-linking of polysaccharides and other intermediates such as phenolic compounds makes the cell wall rigged, which prevent the intruding attempt from pathogens [29]. In watermelon, the reinforcement of the cell wall after pathogen inoculation is thought to be an essential character for breeding the Fusarium wilt resistant lines [2]. CWDEs are practical tools for pathogenic microorganisms to invade host plants [12]. The induction of pectinase and xylanase of *F. oxysporum *f. sp. *ciceris* is positively correlated to the disease development in chickpea [30]. However, ameliorating an individual CWDE shows no significant effect on virulence likely due to functional redundancy [3]. In the present study, the activities of cellulase, pectinase, and xylanase decreased in degenerated variants compared with Fon ST. The phenomenon of losing all arsenals for breaking plant cell wall might reduce the virulence of Fon on watermelon.

Forward and reverse genetics approaches have been used to identify virulence-related factors for pathogenesis. Most of the virulence factors are implicated in fungal cell wall biosynthesis and signal transduction processes. To understand the possible mechanism on the pathogenesis of Fon, we performed qRT-PCR to investigate the correlation between the virulence-related factors and virulence. Gas1, a β-1,3-glucanosyltransferases, a member of glycosylphosphatidylinositol-anchored glycoproteins, is implicated as a virulence-related factor in the pathogenesis of *F. oxysporum* f. sp. *lycopersici* [31]. In addition, Gas1 is required for cell wall biosynthesis and morphogenesis of *F. oxysporum* f. sp. *lycopersici*, and the pathogen with a functional deletion of *gas1* exhibits the structural alterations in the cell wall. The present study showed that the gene expression levels of *gas1* in four variants were all significantly lower than that in their parental culture ST. The data support that *gas1* may be the most functionally important among the tested virulence-related factors in Fon pathogenesis. The virulence-related factors, G-protein β subunit Fgb1 [32], mitochondrial protein Fow1 [33], and pH response transcription factor PacC [34], have been proved to be essential for controlling the virulence of *F. oxysporum*. Our results suggest that collective reduction of required virulence-related factors may involve in the degeneration of the variants. In addition, a reduced activity of CWDEs was detected in the four variants, supporting the hypothesis that the morphological variants in overall have fewer tools than wild-type ST for causing Fusarium wilt on watermelon. The data indicate that sufficient accumulation of virulence-related factors and CWDEs is very important for fully expressing the pathogenicity of Fon.

The possible mechanism of degeneration may be the consequence of genetic polymorphism. Although the cultures tested in this study showed distinct variations, e.g., cultural and pathogenic variability, the genetic diversity between these cultures were not very high because the DNA fingerprinting patterns between the morphologically different cultures by RAPD (random amplification of polymorphic DNA) assays with 48 different random primers (Appendix A) were almost identical. However, RAPD results may not be able to cover all changes in genetic variation such as transposable element. The genetic variation frequently observed in *F. oxysporum* has been attributed to active transposable elements [35]. Other potential mechanisms which change the development and virulence of Fon could also be due to infection of mycovirus or DNA methylation. A chrysovirus Fusarium oxysporum f. sp. dianthi virus 1 (FodV1) has been reported to decrease the ability of *F. oxysporum* f. sp. *dianthi* colonized in their host carnation (*Dianthus caryophyllus*) [36]. The presence of double strand RNA mycovirus is correlated with the changes of pathogenicity and morphology of *F. graminearum* [25]. DNA methylation of plant fungal pathogen is responding to the environment stimuli and incorporate with several biological process including pathogenicity [37]. Therefore, more genome-related research warrants further attention to uncover the mechanism of cultural degeneration in Fon.

Finally, our results of virulence-related genes expression and CWDEs enzyme activities demonstrated significant differences between the variants and their parental ST. These results suggest that expression levels of the tested virulence-related genes and active CWDEs in the variants were insufficient to support their ability to maintain full virulence. Furthermore, the transcript levels of the virulence genes (*fgb1*, *pacC*, *pl1*, *rho1, gas1,* and *wc1*) in the mycelia of the morphological variants, were all significantly lower than in those of their parental culture ST. The data seem to be able to explain why the virulence of the variants was significantly lower than Fon ST by infested-soil method. In addition, it is worth noting that the tested virulence genes and CWDEs may play a role in Fon pathogenesis on watermelon.

It is still far from complete understanding of the underlying roles of individual virulence factors during pathogenesis. Further studies devoted to whole transcriptome profiling of Fon during its pathogenesis may speed up the understandings of the function and mechanism of virulence factors involving pathogenicity individually. In addition, the whole transcriptome profile of Fon shall provide clues to uncovering mechanisms of loss-of-virulence during cultural degeneration.

## Figures and Tables

**Figure 1 jof-06-00382-f001:**
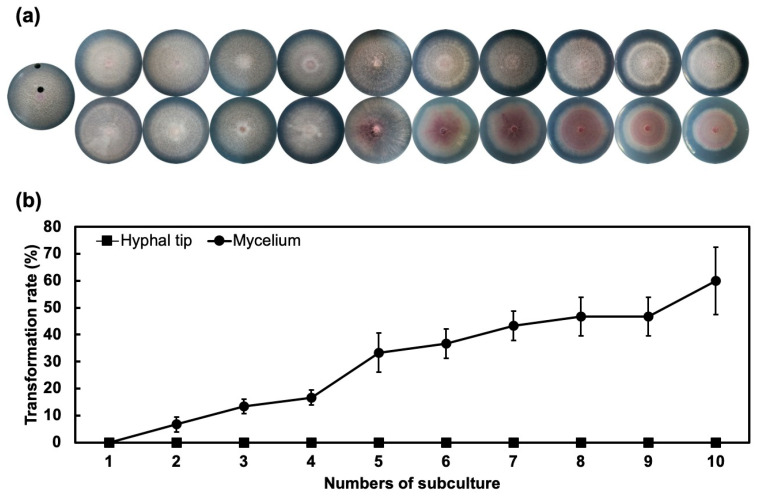
Successive subculture of aged mycelia induces transformation rates of Fon. (**a**) Colonies of Fon grown on half-strength potato dextrose plates at different sequential subcultures (left to right: 1st to 10th subculture) that originated from middle aged mycelia (bottom lane) and fresh hyphal tip (top lane). (**b**) Transformation rates of Fon from different areas of the culture plate. Error bars represent the standard error of three replicated experiments.

**Figure 2 jof-06-00382-f002:**
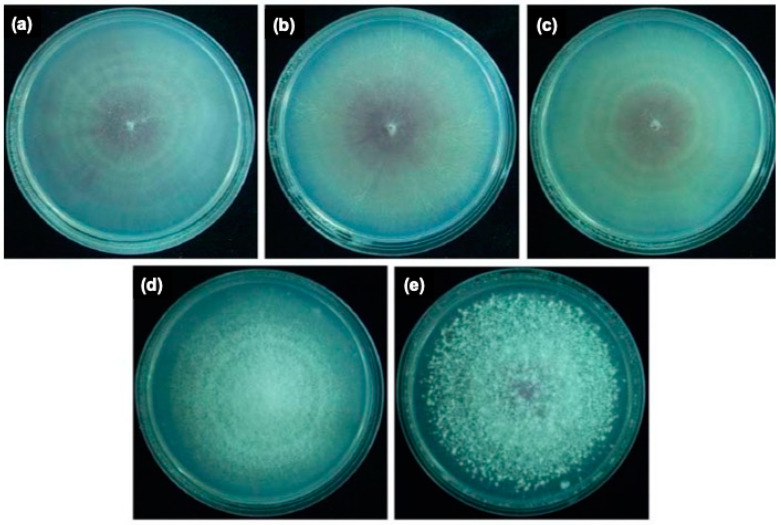
The colony morphology of the watermelon Fusarium wilt pathogen variants PT2, PT3, PT11, MT6, and their parental culture Fon ST. Variants PT2 (**a**), PT3 (**b**), and PT11 (**c**) belong to pionnotal type with slime surface and almost no aerial mycelium formed. Variant MT6 (**d**) belongs to mycelial type with abundant aerial mycelia but lack of sporodochia. Parental ST-H0103 (**e**) belongs to sporodochia type with abundant aerial mycelia and sporodochia.

**Figure 3 jof-06-00382-f003:**
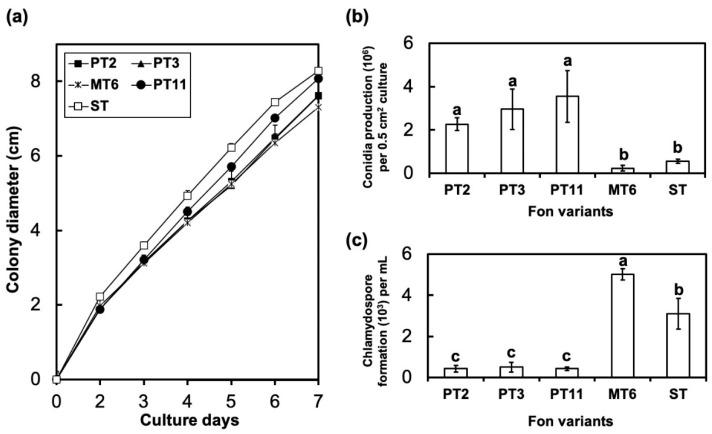
Development of colony diameter (**a**), conidia production (**b**), and chlamydospore formation (**c**) of the watermelon Fusarium wilt pathogen variants PT2, PT3, PT11, MT6, and their parental sporodochial culture Fon-H0103 (ST). (**a**) No significant difference in colony diameter was observed among all Fon cultures. (**b**) The pionnotal variants (PT2, PT3, and PT11) produced the highest amount of conidia, whereas the mycelial variant (MT6) produced the lowest amount. (**c**) The mycelial variant (MT6) produced the highest amount of chlamydospores, and the pionnotal variants (PT2, PT3, and PT11) produced the lowest amount. The values were analyzed by least significant differences (LSD) test and values with different letters indicate significant difference (*p* < 0.05).

**Figure 4 jof-06-00382-f004:**
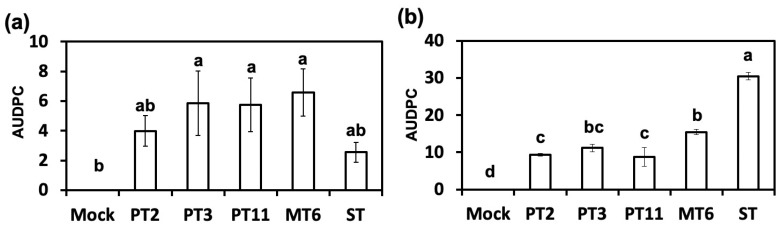
Virulence test of different Fon variants using root-dipping and infested-soil methods. The area under disease progress curve (AUDPC) of different variants through root-dipping (**a**) and infested-soil (**b**) methods was analyzed. Error bars represent the standard error of three replicated experiments. Data with the same letter are not significantly different according to LSD test (*p* < 0.05). Mock: water inoculation control; ST: Fon sporodochial type; PT2, PT3, PT11: pionnotal type; MT6: mycelial type.

**Figure 5 jof-06-00382-f005:**
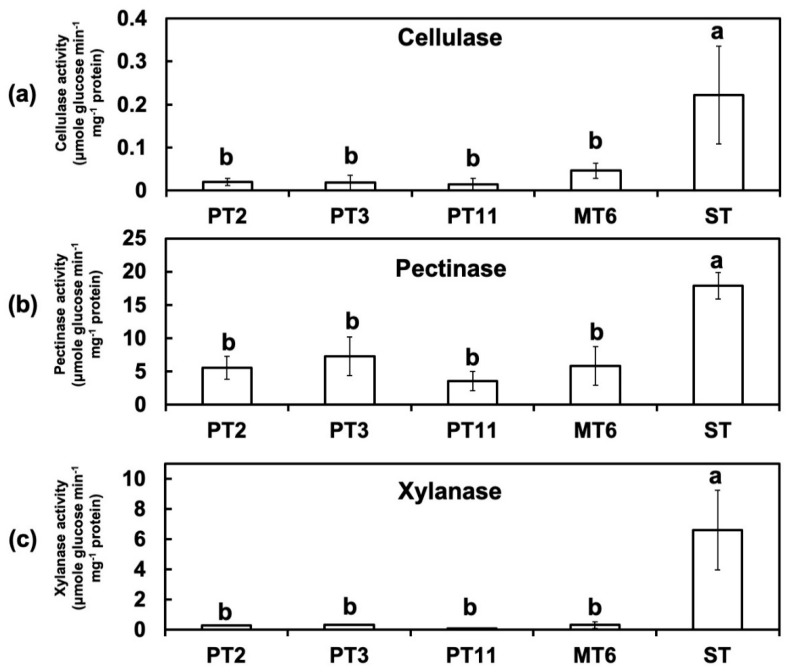
Cellulase, pectinase, and xylanase activities of different variants of Fon. Enzymatic activities of different variants of Fon in cultures with Bilay and Joffe’s medium. Celllulase (**a**), pectinase (**b**), and xylanase (**c**) activities were determined after 7-day incubation. Bars represent the standard error. Data with the same letter are not significantly different according to LSD test (*p* < 0.05). ST: Fon sporodochial type; PT2, PT3, PT11: pionnotal type; MT6: mycelial type.

**Figure 6 jof-06-00382-f006:**
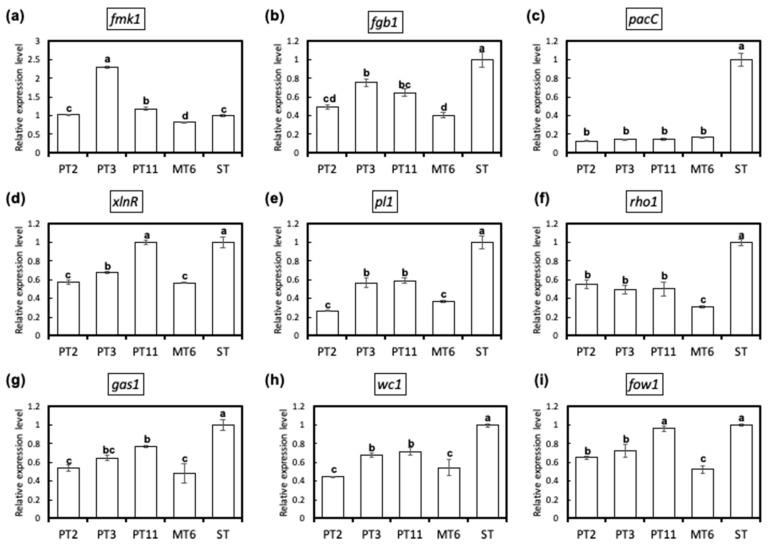
Expression analysis of virulence-related genes by real-time quantitative RT-PCR (qRT-PCR) in the watermelon Fusarium wilt pathogen Fon variants PT2, PT3, PT11, MT6, and their parental sporodochial culture ST-H0103 (ST). Quantification was achieved by normalizing the virulence-related genes *fmk1* (**a**), *fgb1* (**b**), *pacC* (**c**), *xlnR* (**d**), *pl1* (**e**), *rho1* (**f**), *gas1* (**g**), *wc1* (**h**), and *fow1* (**i**) to the endogenous housekeeping gene of elongation factor 1-alpha (EF1α) using the 2^−ΔΔ^ C_T_ method published by Livak and Schmittgen (2001) [22]. The formula 2^−ΔΔ*C*^_T_ was used to calculate the relative transcripts of the virulence-related genes between Fon ST and variants (PT2, PT3, PT11, and MT6). The details of the virulence-related genes are listed in Table 1. Three independent biological replicates for each culture were used for qRT-PCR. The values were analysed by least significant differences (LSD) test and values with different letters indicate significant difference (*p* < 0.05).

**Table 1 jof-06-00382-t001:** Designed primers for virulence genes expression.

Amplicon		Most Homologue			Amplification Primer
Gene Name/bp	Protein Function	Description/Accession Number	Identity (%)	Primer Name	Sequence (5′–3′)
*ef1α*/115	Housekeeping gene used in this study	*Gibberella zeae* PH-1 elongation factor 1-alpha partial mRNA/XM388987	98	ef1aF	TAAGGGTTCCTTCAAGTACG
				ef1aR	GGTGACATAGTAGCGAGGAG
*fgb1*/128	Signal transduction	*F. oxysporum* f. sp. *lycopersici* G protein beta subunit gene (*fgb1*), complete cds/AY219172	98	fgb1F	TGGTGACATGACCTGTATGA
				fgb1R	CACCAGAGATGAAGGTGTTT
*fow1*/142	Signal transduction	*F. oxysporum* plasmid pWB60Sl FOW1 gene (*fow1*) for putative mitochondrial carrier protein, complete cds/AB078975	100	fow1F	GAGATCACCAAGCACAAGAT
				fow1R	AGTTGGTCATCTGTCGGATA
*fmk1*/147	Mitogen-activate protein kinase	*F. oxysporum* f. sp. *lycopersici* mitogen-activate protein kinase gene (*fmk1*), complete cds/AF286533.1	99	fmk1F	ACATTCGATCGCTCCCCTTC
				fmk1R	ATGGGTGCTTCAGAGCTTCC
*gas1*/119	Fungal cell wall biosynthesis	*F. oxysporum* beta-1,3-glucanosyltransferase gene (*gas1*), complete cds/AY884608	98	gas1F	GTGAGACCAACGACAAGACT
				gas1R	GATAGCGTAAGTTCGGATGA
*pacC*/124	Negative regulator of pH signaling	*F. oxysporum* pH transcription factor gene (*pacC*), complete cds/AY125958	95	pacCF	CTGGTCCTACTCCTAGCAAA
				pacCR	TAGATGGTGTCTTGCATCTG
*pl1*/113	Pectin degrading enzyme	*F. oxysporum* f. sp. *lycopersici* pectate lyase gene (*pl1*), complete cds/AF080485	98	pl1F	TTCATTCCCTACTGCTGTTC
				pl1R	TAGCATCCTTCTCACCAGTT
*xlnR*/133	Transcription activator of xylanolytic system	*F. oxysporum* f. sp. *lycopersici* XlnR gene (*xlnR*), complete cds/EF057395.1	99	xlnRF	TGGACAAAGGACGGAACAGG
				xlnRR	CACCCGAAAGAGAGTGTCCC
*rho1*/104	Fungal cell wall biosynthesis	*F. oxysporum* small GTPase-binding protein gene (*rho1*), complete cds/AY884607	99	rho1F	GATACGACCAGAAGACCATC
				rho1R	GTACTTGTAGGCAGCGATCT
*wc1*/129	Photoreceptor	*F. oxysporum* f. sp. *lycopersici* white collar 1 gene (*wc1*), complete cds/EU327188	99	wc1F	TCAACAGTGTGTCATCCATC
				wc1R	GCTTTCGAACCAAGTGTAAC

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
