# Peer review of "Degenerated Virulence and Irregular Development of Fusarium oxysporum f. sp. niveum Induced by Successive Subculture"

_jof, 2020, doi:10.3390/jof6040382_

Round 1
Reviewer 1 Report
This manuscript mainly reported the degeneration of developmental phenotype and virulence in Fusarium oxysporum f. sp. niveum (Fon) during its successive cultivation. The content itself could be of potential interest as it might expand our understanding of fungal pathogenesis and other biological characteristics. However, i think some of the details need to be improved.
- The authors declaimed in the manuscript that the ST and other variants had no significant differences in virulence in the root-dipping inoculation experiment. However, this was not consistent with the descriptions on the Abstract part.
- Fungus degeneration is a commonly phenomenon in successive subculture of aged mycelia, even though their rate of occurrence is uncertain. In this article, the authors should list out how many culture dishes were used in each subculture procedure.
- I think the experiments of RAPD for genetic background analysis could be moved into the text part in order to exclude the possibility of strains contamination.
- My main concern is that there has no direct evidence that the cultural degeneration could lead to the loss of pathogen virulence-related factors in the early stage of Fon infection process. After all, some histological and ultrastructural analysis regarding to microscopic morphology and infection process of the pathogen fungus were needed.
- On the other hand, i think it is more reasonable to select more than one degenerated mycelium type variant in order to eliminate the error of single strain to the maximum extent.
- In our experience, fungus degeneration might be associated with other factors, such as mycovirus infection, and epigenetic modification. Thus, focusing on these areas, if possible, may yield some unexpected results.
- Some language problems in the text also need to be strengthened.
Author Response
Thank you for your generous help, we have attached a pdf version revise with track mark.
- The authors declaimed in the manuscript that the ST and other variants had no significant differences in virulence in the root-dipping inoculation experiment. However, this was not consistent with the descriptions on the Abstract part.
Respond1: We have corrected the descriptions in the Abstract content (Line 30). - Fungus degeneration is a commonly phenomenon in successive subculture of aged mycelia, even though their rate of occurrence is uncertain. In this article, the authors should list out how many culture dishes were used in each subculture procedure.
Respond2: In the supplement Figure 2, we have listed all the plates in a set of experiment. We added the details in the materials and methods as requested (Line 182-184). Ten culture dishes were used in each subculture procedure. This experiment was repeated three times. - I think the experiments of RAPD for genetic background analysis could be moved into the text part in order to exclude the possibility of strains contamination.
Respond3: We agreed that RAPD can be used for excluding the strains contamination. However, our RAPD result was to compare the structure variations of genome between ST and variants. In order to exclude the possibility of strains contamination, we used single spore isolation for maintaining strains to for avoid the unexpected fungi. - My main concern is that there has no direct evidence that the cultural degeneration could lead to the loss of pathogen virulence-related factors in the early stage of Fon infection process. After all, some histological and ultrastructural analysis regarding to microscopic morphology and infection process of the pathogen fungus were needed.
Respond4: Thanks for the advice and concern. Unfortunately, we have no such evidence to show the direct contact of pathogens and watermelon. We will work on that but it will take some time to get analysis done. We think the results of three times root-dipping inoculation and three times infested-soil inoculation provide sufficient evidences to show the virulence loss in degenerated variants. - On the other hand, i think it is more reasonable to select more than one degenerated mycelium type variant in order to eliminate the error of single strain to the maximum extent.
Respond5: The occurrence of mycelium type is lower than pionnotal type. Unfortunately, we only have one variant of mycelium type for the following experiments. - In our experience, fungus degeneration might be associated with other factors, such as mycovirus infection, and epigenetic modification. Thus, focusing on these areas, if possible, may yield some unexpected results.
Respond6: We agreed with the reviewer’s point. It is possible other factors as mentioned by this reviewer could be associated with degeneration. More discussions were added (Lines 568-578). We will focus on this area in the future. - Some language problems in the text also need to be strengthened.
Respond7: We have sent the content to our senior colleagues, Professor KR Chung and Professor CJ Chang, to check the language problem. Professor KR Chung is currently an editor in Journal of Fungi. Professor CJ Chang is currently Professor Emeritus at University of Georgia, USA.
Reviewer 2 Report
In this article the Authors addressed a phenomenon often occurring in filamentous fungi as a consequence of repeated culture on artificial media, i.e. loss of pathogenicity, growth reduction and morphological changes (colony saltations, reduction of vegetative growth, increase or loss of sporulation ability). This phenomenon is collectively referred to as mycelium degeneration.The Authors studied Fusarium oxysporum f. sp. niveum, the causal agent of Fusarium wilt of watermelon, as a model system. To this aim they compared a wild strain of the fungus and diverse types of degenarated variants. The novelty of this study is that the Authors found a correlation between the mycelim degeneration, including loss of pathogenicity, and the activity level of some cell wall degrading enzymes as well as the expression of genes related to the virulence in fungal pathogens. They concluded that these cell wall degrading enzymes are virulence factors.
In general the experimental design is appropriate and the graphical presentation of results is satisfactory.
In my opinion major limits of this article are as it follows:
- the English style needs to be substantially improved; some sentences have to be completely rephrased. I suggest the Authors to ask the help of a mother-tongue colleague for a critical review before resubmitting the paper;
- the Authors’ statement that the degerative features they observed were irreversible should be proved with experimental results that have to be included in the article;
- the Discussion is too speculative and considers that the physiological and genetic alterations the Authors observed affected the virulence. The explanation the Authors provided to justify the contrasting results between two different inoculation methods (direct, by root dipping, and indirect through the soil) are not convincing and should be better supported by the literature. In this respect, the Authors did not consider as an alternative hypothesis that differences between the wild parental strain and the variants they higlighted could have also affected the fitness of variants, as suggested by the differences in virulence observed using two different methods of inoculation. In my opinion the Authors can only hypothesize that cell wall degrading enzymes are virulence factors in this pathosystem.
- A specific criticism concerns the use of RAPDs to detect transposable elements and to exclude their role in the origin of saltations. I think that RAPDs are not a suitable tool to detect
Other minor observations and suggestions are included as notes in the text (see attached file)

Author Response
- the English style needs to be substantially improved; some sentences have to be completely rephrased. I suggest the Authors to ask the help of a mother-tongue colleague for a critical review before resubmitting the paper;
Response 1: We have asked our senior Professor to help us improve the English style of this article. Professor KR Chung is currently an editor in Journal of Fungi. Professor CJ Chang is currently Professor Emeritus at University of Georgia, USA. - the Authors’ statement that the degerative features they observed were irreversible should be proved with experimental results included in the article;
Response 2: The degeneration of phenotype has been reported before. We have cited the literature in this article (Line 443-445). Also, each variant was preserved with single spore that prevent further contaminations. We also observed that cultural phenotype of those variants were the same while isolated from their infested-soil (written in the context) (Line 445-447). - the Discussion is too speculative and considers that the physiological and genetic alterations the Authors observed affected the virulence. The explanation the Authors provided to justify the contrasting results between two different inoculation methods (direct, by root dipping, and indirect through the soil) are not convincing and should be better supported by the literature. In this respect, the Authors did not considered as an alternative hypothesis that differences between the wild parental strain and the variants they evidentiated could have also affected the fitness of variants, as suggested by the differences in virulence observed using two different methods of inoculation. In my opinion the Authors can only hypothesize that cell wall degrading enzymes are virulence factors in this pathosystem.
Response 3: Thanks for the reviewer’s advices, we have changed the content and added more references. We have changed some discussion in our context (Line 455-456 and Line 463-464). - A specific criticism concerns the use of RAPDs to detect transposable elements and to exclude their role in the origin of saltations. I think that RAPDs are not a suitable tool to detect transposons.
Response 4: Thanks for the advice. RAPDs are actually not suitable to identify the transposable element. The reason we applied RAPDs was to identify the polymorphisms of variants’ genome. But this cannot rule out the possibility of transposable element as well as DNA methylation. We have cited several literatures to address the issues (Line 541-578). - Other minor observations and suggestion are included as notes in the text (see attached file)
Response 5: We have corrected the content as reviewer’s request. Many thanks for reviewer’s help.
Round 2
Reviewer 1 Report
Prior comments have been addressed or explained satisfactorily.